# Assessing the Relational Abilities of Large Language Models and Large Reasoning Models

**DOI:** 10.3390/bs16010045

**Published:** 2025-12-25

**Authors:** Matthias Raemaekers, Martin Finn, Jan De Houwer

**Affiliations:** Department of Experimental Clinical and Health Psychology, Ghent University, 9000 Ghent, Belgium; martin.finn@ugent.be (M.F.); jan.dehouwer@ugent.be (J.D.H.)

**Keywords:** large language models, reasoning models, relational reasoning, relational abilities index, transformation of function

## Abstract

We assessed the relational abilities of two state-of-the-art large language models (LLMs) and two large reasoning models (LRMs) using a new battery of several thousand syllogistic problems, similar to those used in behavior-analytic tasks for relational abilities. To probe the models’ general (as opposed to task- or domain-specific) abilities, the problems involved multiple relations (sameness, difference, comparison, hierarchy, analogy, temporal and deictic), specified between randomly selected nonwords and varied in terms of complexity (number of premises, inclusion of irrelevant premises) and format (valid or invalid conclusion prompted). We also tested transformations of stimulus function. Our results show that the models generally performed well in this new task battery. The models did show some variability across different relations and were to a limited extent affected by task variations. Model performance was, however, robust against the randomization of premise order in a replication study. Our research provides a new framework for testing a core aspect of intellectual (i.e., relational) abilities in artificial systems; we discuss the implications of this and future research directions.

## 1. Introduction

Large language models (LLMs, e.g., GPT4, [53]; LlaMa 3, [21]) and large reasoning models (LRMs, e.g., Open AI o1, [54]; DeepSeek R1, [31]) have taken the world by storm. Their ability to produce human-like text, answer questions and solve problems (and much more) has inspired much debate about how intelligent such systems are, and to what extent this aligns with human intelligence. While many have taken their performance as an indication that superhuman (or even general) intelligence is within reach (e.g., [4]), such conclusions are nuanced by reports of those same systems failing on tasks that are trivial for the average human (e.g., [8]; [47]; [5]). The debate about these systems’ level of intelligence is complicated by a range of issues. First, there is little consensus about a definition of human intelligence (e.g., [65]; [40]), let alone artificial or general intelligence (e.g., [67]; [12]; [51]), which leads to inconsistencies in how different perspectives interpret the same findings. Second, the fact that these language models are so fluent at producing human-like text, and that they are often tested on tasks designed for humans (e.g., the bar exam, intelligence tests) brings the risk of anthropomorphizing these systems ([17]; [62]), attributing human-like understanding, intelligence and even goals, purpose and sentience to them (often referred to as the ELIZA-effect). However, the alignment between human and artificial system’s responses does not a priori reflect alignment of abilities or the mechanism underlying them ([47]; [43]). It is important to see these models for what they are—large neural network models that are trained to predict the next token in a text—and evaluate them as such ([47]). Finally, because these systems are trained on massive datasets, it is not uncommon that some of the test items used to evaluate them were present in the training data and that performance measures are biased as a result. This further highlights the need for systematic and rigorous testing of state-of-the-art LLMs and LRMs ([9]; [25]; [64]; [72]; [41]; [71]). In this research, we take an approach towards evaluating LLM and LRM capacities inspired by behavior-analytic research on relational abilities. Doing so allows us to avoid getting caught up in philosophical debates about the nature of general intelligence, yet still obtain a measure closely related to it (i.e., do these systems demonstrate a behavior that is believed to be a hallmark of intelligence; [16]).

Researchers from different fields in psychological research have emphasized the importance of relational abilities as a cornerstone of human cognition (e.g., [33]; [57]; [32]; [28]; [35]) and intelligence more generally (e.g., [11]; [16]; [49]). Tests of relational abilities are part of many modern intelligence tests (i.e., tasks like matrix reasoning, verbal analogies, etc.; e.g., [61]; [39]; [69]), and some have been proposed to be a proxy-measure of intelligence (e.g., [14]; [46]). Interestingly, different types (or levels of complexity) of relational abilities can be assessed and compared in different types of systems, be that of nonhuman animals, humans or artificially intelligent systems. To see how, we first need to describe the behavior-analytic perspective on relational abilities in more detail.

Relational frame theory (RFT, [33]) provides a behavior-analytic definition of relational responding—the ability to respond to one event or stimulus in terms of its relation to another. This definition encompasses both responding to formal relations between events (e.g., choose the larger of two stimuli; referred to as non-arbitrarily applicable responding or NAARR) and responding to symbolic relations (referred to as arbitrarily applicable relational responding or AARR). In the latter case, relations can be arbitrarily applied to stimuli (i.e., regardless of formal properties, e.g., selecting a smaller, more valuable €1 coin over the larger, less valuable €0.50 coin) if particular contextual cues are present. Relational contextual cues signal the relationship between the stimuli (sameness, difference, hierarchy, and so on), whereas functional contextual cues indicate what stimulus function is related (e.g., color, value, threat). AARR has three core characteristics: (1) mutual entailment refers to the bidirectionality of relations (if A is related to B, B is also related to A); (2) combinatorial entailment or transitivity (if A is related to B and B is related to C, one can derive the relation between A and C); and (3) the transformation of function (i.e., if a neutral stimulus A is related to a stimulus B with a reinforcement value, the reinforcement value of A will be transformed according to the specified relation and one will respond to A as if it has that function). The definition of relational responding is in line with related constructs studied in cognitive psychology, such as relational reasoning ([1]) or analogical reasoning ([34]; [28]). We would like to emphasize the fact that the definition encompasses both NAARR (i.e., responding to formal stimulus relations) and AARR (responding to contextually defined stimulus relations, regardless of formal stimulus relations). Its broad scope allows us to compare different systems (nonhuman animals or other organisms, humans, machines, …) on a vast array of behaviors, from animals responding to simple, formal sameness and difference relations (e.g., a relational matching-to-sample task or transitive inference task; [57]) to humans or machines solving complex analogies or metaphors (e.g., [68]; [63]).

Many tests have been developed to measure relational abilities (e.g., [58]; [29]; [7]; [1]; [14]). In this research, we will use a measure developed within the behavior-analytic literature, referred to as the relational abilities index (RAI, [14]). The procedure is similar to syllogistic reasoning tasks in that it presents the participant (or language model) with a number of relational premises involving nonwords (e.g., “COR is the same as WUG and WUG is different from LOM”) and a (or in some versions, multiple) conclusion(s) to respond to (e.g., “Is COR the same as LOM”?). Syllogistic reasoning has long been considered a hallmark of human intelligence (e.g., by Aristotle) and is studied in various research areas such as behavioral psychology (e.g., [48]; [14]); cognitive psychology (e.g., [30]; [66]) and neurology (e.g., [29]), as well as computer science (e.g., [36]; [2]; [22]; [56]). Traditionally, syllogistic reasoning tasks make use of realistic stimuli (e.g., names of people, animals, etc.) and involve either comparative relations (e.g., “Ben is smaller than Tom, Tom is bigger than Lisa”, see [29], for an example) or (logical) reasoning about object categories and features (e.g., category-based induction problems, e.g., “All birds have wings; Penguin is a bird; Therefore penguins have wings”, see [22], for an example). The RAI, on the other hand, involves multiple relations (sameness, difference, opposition, more or less than, hierarchical relations, temporal relations, deictic relations and analogies; see [18], for a detailed discussion) and uses arbitrarily chosen stimuli (i.e., nonwords). As such, it aims to capture the generalized (abstract) ability to respond relationally in an arbitrarily applied fashion (i.e., regardless of formal stimulus properties) and may provide a broader survey of relational reasoning abilities than similar measures do. The procedure is well-suited for testing language models’ relational abilities also. The fact that they are natural language reasoning problems makes them readily applicable in LLMs and LRMs and the use of nonwords reduces the likelihood that the models encountered identical problems in training. While the models may still benefit from structurally similar problems that are almost certainly present in training data, syllogistic reasoning problems can be manipulated in many ways to create different variations in the same problems (e.g., increasing the number of premises, including irrelevant premises, manipulating the order of premises, changing the response format to multiple choice) so as to further reduce the likelihood of contaminated results.

For this study, we created a large battery of several thousand syllogistic reasoning problems inspired by the RAI. We assessed a multitude of relations (sameness, difference, opposition, more than, less than, hierarchical relations, temporal relations, deictic relations and analogies). The task was created making use of relational derivation tables (essentially look-up tables that store the reversal and transitive combinations of different combinations of relations allowing for automated derivation of relations; [59][note 1]). It represents a significant increase in size relative to similar tasks, and to our knowledge, this is the largest and most varied RAI-inspired task to this date (previous versions ranged from 55, [14], to 67 trials, [15], [18]). In addition to relations typically tested in the RAI, we also included test trials that specifically assessed transformation of function. RFT considers this to be an important characteristic of relational responding, but it is currently not included in the RAI. We manipulated problem complexity by varying the number of premises (i.e., one to five relevant premises, with an optional irrelevant premise added to the problem). We also created further problem variations by manipulating whether a valid or invalid conclusion was prompted and by generating multiple instances of each unique problem using different nonwords in the premises, to get a reliable estimate of model performance. We describe the task in more detail below.

## 2. Materials and Methods

### 2.1. Syllogistic Reasoning Problems

We constructed a large battery of syllogistic relational reasoning problems (N = 1364 unique problems) that included a wide variety of relations and varied in structure and complexity. Similarly to the RAI, for descriptive purposes, we group the trials into blocks based on the relations they involve, the complexity of the relational arrangement, and the requirement to transform stimulus functions: (a) sameness and difference (N = 152), (b) sameness and opposition (N = 488), (c) more than and less than (N = 72), (d) before and after (N = 72), (e) hierarchy (i.e., ‘contains’ and ‘is part of’; N = 72), (f) deictic (I-you, here-there, now-then; N = 48), (g) analogy (N = 52) and (h) transformation of function (N = 408). The different trial numbers for the different relations were the result of the functioning of the generator function. For a number of specified relations, this function finds all valid combinations (i.e., the combination allows for valid derivation of a new relation) of the relations for a given number of premises (e.g., for two premises with sameness or difference relations, it could be sameness twice, or a one of two combinations of sameness and difference). For those combinations of relations, it then creates relational syllogistic problems with one relation per premise. Relations vary in terms of how they can be combined with other relations to allow valid derivations. For instance, two (or more) opposition relations can be specified between three (or more) relata, allowing one to derive novel transitive relations (e.g., A opposite to B and B opposite to C, so A is the same as C), whereas combinations of two or more difference relations does lead to valid derivations (if A is different from B and B is different from C, one cannot derive the relation between A and C with certainty). As a result, for some combinations of relations (e.g., same and different), fewer unique trials could be generated, leading to different numbers of trials between the relations. Each problem consisted of between one and six relational premises (e.g., ‘A is the same as B. B is the same as C’) and then prompted a conclusion, to which a binary ‘yes’ or ‘no’ response was required. We systematically manipulated problem complexity (varying number of premises and inclusion of an irrelevant premise) and the correct response (prompted syllogism conclusion was valid on half of trials, invalid on the other half). For deictic relations, we also manipulated whether the deictic dimension was to be reversed or not (see [48]). Ten variants (each with different non-words as the relata) of each unique problem were created, adding up to a total of 13,640 problems (full set available on the Open Science Framework[note 2]) on which the LLMs and LRMs were assessed. To reduce the likelihood of prior experience with them in this context, the non-words used as relata were randomly selected from a pool of three-letter syllables, chosen to not be included in openly available versions of the RAI and to as much as possible not be parts of common words. Randomization of the non-words in different problems was constant across the different models. Below, we describe the main variables (beyond the relation involved) that we manipulated in the problems. Example problems are illustrated in Figure 1.

#### 2.1.1. Number of Premises and Irrelevant Premises

For all relations except deictic and analogy relations (which always had two premises with an optional irrelevant premise), we manipulated the number of premises in the problems from one to five premises. Problems with only one premise can be thought of as assessing mutual entailment or reversal of a relation (e.g., “A is the same as B, is B the same as A?”, see Figure 1A), whereas problems with two or more premises assessed understanding of transitivity of relations (e.g., “A is opposite to B, B is opposite to C, and C is the same as D. Is D the same as A?”, Figure 1B). For each level of complexity, we also included a variant where an additional, irrelevant premise was added before prompting the conclusion (e.g., Figure 1C,F,I,M). That is, this premise referred to one of the related nonwords, but was not relevant for the prompted derivation (e.g., “A is the same as B, B is opposite of C and C is the same as D. Is A the opposite of C?”).

#### 2.1.2. Conclusion Validity

We also manipulated the validity of the prompted conclusion. That is, half of the problems had an invalid conclusion prompted (e.g., “A is the same as B. Is B different from A?”, e.g., Figure 1B), while the other half had a valid conclusion prompted (e.g., “A is the opposite of B and B is the opposite of C. Is C the same as A?”; e.g., Figure 1D). This manipulation allowed us to investigate whether the models show signs of affirmation bias (i.e., better performance when answering yes). Such biases could, for example, arise due to problems with correct conclusions being more prevalent in training than those with invalid conclusions, or because of the extensive finetuning process to tailor the models’ responses to human preferences.

#### 2.1.3. Deictic Relations

The problems involving deictic relations had a slightly different structure to those involving other relations (see Figure 1G–J). RFT describes three types of deictic relations ([48]): interpersonal relations (I–you), temporal relations (now–then) and spatial relations (here–there)). RFT-inspired tests of this ability (e.g., [48]; [19]) combine these relations (e.g., I am here now, you were there then. Am I here now?’). For the current test battery, we separated the three relations. One set of trials assessed spatial relations (‘A is here. B is there. Is B here?’), another assessed purely temporal relations (e.g., ‘A is now, B is tomorrow. Is B tomorrow?’) and a final set assessed interpersonal relations (e.g., ‘I think A, you think B. Do I think A?’). Instead of different nonwords, different objects, events and thoughts were included in the various iterations of each problem. Similarly to the other relations, we created trial variations where an irrelevant premise was included (Figure 1I), and where the prompted conclusion was invalid (as well as a combination of both). We also included a variant that prompted a reversal of the deictic dimension (e.g., ‘If I were you and you were me, would I …?’ or ‘If here were there and there were here, would A be here?’, Figure 1J).

#### 2.1.4. Analogy

For all relations except deictic relations, we included problems that required analogical reasoning (see Figure 1D–F). These problems provided two premises (e.g., “A is opposite to B, B is opposite to C.”) and then prompted a comparison of the relation in the first premise to that in the second premise (Figure 1D). Here too, a version with an incorrect analogy prompt (e.g., “A same as B and B same as C. Is A to B different from B to C?”, Figure 1E), and a version with a third, irrelevant premise (i.e., not involved in the analogy prompt, Figure 1F) were included.

#### 2.1.5. Transformation of Function

To assess whether LLMs and LRMs demonstrate the ability to transform psychological functions, we also created a set of problems that applied relations to semantic functions of nonwords (see Figure 1K–M). In these problems, a non-word is assigned a meaning, then related to other nonwords in the same relational premises, and then the meaning of one of the related nonwords was queried (e.g., “AGU means ‘yes’. AGU has the same meaning as BUR. BUR has the opposite meaning of LOP. Does LOP have the same meaning as ‘yes’?”, Figure 1K). For sameness, difference and opposition relations, the nonwords were related to existing words (e.g., “ARU has the same meaning as ‘cat’.”), whereas for more than, less than, before and after relations, nonwords were assigned a value (e.g., “ARU is worth €5”) or time (e.g., “ARU is at 4 p.m.”). Variations in these problems again included manipulating the validity of the presented conclusion (Figure 1L) and adding irrelevant premises to the problem (Figure 1M). While this is a limited form of transformation of function, we do consider it to be a transformation of semantic function via sameness, difference, opposition, comparative and temporal relations. In that sense, we go beyond what existing versions of the RAI measure and at least have some measure of a crucial aspect of relational responding abilities.

#### 2.1.6. Replication with Randomized Order of Premises

As a final test of the generality of the models’ ability to solve relational syllogistic problems, we replicated the full task battery as described above (i.e., 13,640 problems) but randomized the order of premises in each problem. That is, for each problem consisting of two or more premises (including irrelevant premises and premises specifying a function), the premises where no longer presented in a linear fashion (i.e., “A related to B, B related to C, C related to …”), but in a randomized order (e.g., A related to B, C related to D, B related to C). The (randomized) premises were still followed by a prompted conclusion requiring a yes or no response from the models. While randomizing the order of premises might also increase the complexity of the problems for humans, in principle, the relational complexity (i.e., the derivations to be made) remains the same. Therefore, reduced performance in this replication would hint at a lack of generalized ability for relational responding.

### 2.2. Models and Inference

We tested a sample of two state-of-the-art LLMs and two LRMs. This sample was chosen to include LLMs and LRMs varying in size, architecture and training. Although our sample was too limited to make general claims about LLMs, LRMs, or the difference in performance between the two, including a variety of models did allows us to assess relational behavior in both type of systems. Given the seminal nature of our study, we restricted the sample because of financial, environmental and practical concerns[note 3]. All models were accessed using the Together AI API service[note 4]. We describe them briefly below (summary in Table 1). For more technical descriptions, see the referenced publications.

We tested two models from the Llama 3 collection of LLMs developed by Meta AI ([21]). Llama 3.1 405B, which has 405 billion parameters and a decoder-only transformer architecture, and Llama 3.3 70B, which has 70B parameters and has received further training (reinforcement learning, RL, from human feedback) to fine-tune it to human preferences. We also tested two recent models of the GPT-model family ([52]): GPT OSS 20B and GPT OSS 120B. These are so-called open-weight models that are trained using reinforcement learning from human feedback and other techniques derived from past Open AI models, on only text data ([52]). Both have a mixture-of-experts architecture (MoE, [37]). GPT OSS 20B has 21 billion parameters with 3.6 billion active parameters, GPT OSS 120B has 117 billion parameters with 5.1 billion active parameters. Based on the results of a pilot optimization study (see Appendix A), we chose to test the models in a zero-shot setting (i.e., no examples, information or additional prompts provided in the context; a system prompt was provided: “Answer with yes or no only, no punctuation is needed. Don’t put a space before your response.”), and with the temperature parameter set to 0.75. For all other parameters, the default settings were used (see Together AI documentation).

## 3. Results

### 3.1. Overall Task Performance

Figure 2 (and Table 2) shows the models’ overall performance by blocks (i.e., relations, including analogy and transformation of function separately). As per the preregistered analytic strategy, we limit ourselves to descriptive statistics here to keep the results digestible. Binomial Generalized Linear Model analyses are summarized in Appendix C. Our results demonstrated that the LRMs (GPT OSS 20B and GPT OSS 120B) mostly outperform the LLMs (LlaMa 3.1 405B and Llama 3.3 70B; except for deictic relations where LlaMa 3.1 405B outperformed the other models slightly). The larger models (more parameters) also tended to outperform their smaller counterparts. While the models generally reached a high level of performance (GPT models correctly solved 83% or more of the problems and reached maximum or near-maximum scores in most blocks), there was quite some variability in performance across the different blocks (more so in LlaMa models). Performance dropped notably in the analogy block, even for the GPT models (GPT OSS 20B: 83.27%; GPT OSS 120B: 83.85%), but more so for the LLMs (LlaMa 3.1 405B: 55.19%; LlaMa 3.3 70B: 55.31%). For the LLMs, performance also dropped significantly in the same–opposite block (LlaMa 3.1 405B: 61.25%; LlaMa 3.3 70B: 55.18%) and the transformation of function block (LlaMa 3.1 405B: 82.06%; LlaMa 3.3 70B: 69.04%), and to a lesser extent in the same–different block (LlaMa 3.1 405B: 83.16%; LlaMa 3.3 70B: 70.59%).

### 3.2. Effect of Problem Complexity

For all blocks except the deictic and analogy blocks, we varied the number of relational premises in each problem from one to five. For comparative (more than, less than) and temporal relations (before, after), all models were remarkably robust to this manipulation and performed at near-maximum level for all levels of complexity. For the other relations, however, we did observe that performance dropped increasingly with more premises. In the same–different and same–opposite blocks, this drop in performance was more pronounced for the LlaMa models (which dropped from maximum down to chance-level) than for the GPT models (which dropped by about ten percentage points). For hierarchical relations, we observed a reversed pattern, where the models’ performance dropped by up to 20% for problems with three premises, but then recovered slightly for more complex problems. These results are illustrated in Section B.1 Figure A2 and described in Table A5.

### 3.3. Effect of Prompt Validity

We also manipulated whether a valid or invalid conclusion was prompted and whether an irrelevant premise was included, to test the models’ sensitivity to variations in problem presentation. While the effect of these variations was more difficult to interpret, we can say that GPT models were less affected by them than were the LlaMa models (see Section B.2 Figure A3 and Table A6 for detailed results). For the same–different, more than–less than and before–after blocks, all models performed consistently (at a high level) across the problem variations. For same–opposite relations, the GPT models were relatively unperturbed by the problem variations and were even more accurate when an incorrect conclusion was prompted (10–15% more accurate). The LlaMa models showed the reversed patterns, being significantly less accurate when an irrelevant premise was included in the problem and with performance dropping even more when an invalid conclusion was prompted. Finally, for hierarchical relations, both the GPT and LlaMa models’ performance dropped slightly (between ten and thirty percent) when an invalid conclusion was prompted, but the addition of an irrelevant premise did not noticeably affect performance. We discuss the effect of problem variations for deictic relations, analogy and transformation of function separately below.

### 3.4. Deictic Relations Performance

We describe the results for deictic relations separately, as these problems were slightly different from the other relations. Trials involved three types of deictic relations: interpersonal, temporal and spatial. All models were slightly less accurate on problems involving interpersonal relations than on trials involving spatial and temporal relations, on which they reached near-maximum performance (see Section B.2 Figure A3 and Table A7). In addition to manipulating the inclusion of an irrelevant premise and the validity of the prompted conclusion, we also included trials where the deictic dimension was reversed. While the models were only modestly affected by prompting an incorrect conclusion, performance dropped when a reversal was included, and more so when this was combined with an invalid conclusion or irrelevant premise. Most models were affected by trial variations in similar ways, but it is noteworthy that LlaMa 405B appeared to be significantly more robust to these manipulations, with its performance remaining largely stable or even increasing (relative to the regular prompt), and likely as a result outperforming the other models on deictic responding with 92.17% accuracy).

### 3.5. Analogy Performance

For all relations except deictic relations, we probed analogical reasoning abilities by providing two relational premises and asking whether the relations in the two premises were the same or not. Here too, we included variants with an invalid conclusion and with irrelevant premises. Across problem variants, models were more accurate solving analogies involving sameness, difference and opposition relations than those involving comparative (more than, less than), temporal (before, after) or hierarchical relations (see Section B.3 Figure A4 and Table A8). Still, in the same–different and same–opposite blocks, LlaMa 3.1 405B was significantly less accurate on problems where an incorrect conclusion was prompted (performance dropped from around eighty to below fifty percent). The smaller LlaMa 3.3 70B model only showed this sensitivity in the same–opposite block, while the GPT-models were relatively unaffected by problem variations in these blocks. In the blocks involving comparative, temporal and hierarchical relations, performance was markedly lower. The LlaMa models scored below 25% accuracy, while the GPT models reached about 50% accuracy. Performance on problems involving temporal and hierarchical relations was slightly higher, with LlaMa models reaching about 30% accuracy and GPT models reaching up to 75% accuracy. Interestingly, in the latter three blocks, performance appeared to increase when an invalid conclusion was prompted, or when an irrelevant premise was included.

### 3.6. Transformation of Function Performance

For sameness, difference, opposition, comparison and temporal relations, we also included trials that probed the models’ ability to transform stimulus functions (nonword meaning). These problems again included variants with invalid conclusions and irrelevant premises. The GPT models were both highly accurate for all relations (near-maximum performance, GPT OSS 20B 96.74% on average, GPT OSS 120B 99.24% on average). The LlaMa models were slightly less accurate (LlaMa 3.3 70B was 69.04% accurate on average, LlaMa 3.1 405B was 82.06% accurate on average), but this drop in performance was limited to same–different and same–opposite relations. For those relations, both LlaMa models’ performance decreased with an increasing number of premises and with invalid conclusions prompted (see Section B.4 Figure A5 and Table A9 and Table A10).

### 3.7. Replication with Randomized Order of Relational Premises

As a final test of the robustness of our results, we conducted a replication study in which we recreated the full task battery, but with the order of relational premises randomized for each problem. The results of this replication are illustrated in Figure 3 and summarized in Table 3. As can be seen from comparing Figure 1 and Figure 3, the models generally performed at about the same level in the replication study as in the original task. For some models, in particular blocks, performance dropped significantly, while for others, it even improved slightly. The most notable drop in performance was for GPT OSS 120B. While still performing at a high level, it lost around five percentage points in the same–different (91.84%), same–opposite (88.81%), comparison (94.03%), temporal (94.31%), hierarchy (92.92%) and transformation of function blocks (92.48%), in which it performed at near-maximum level in the original task (see Table 2). The smaller GPT OSS 20B model was more robust to the randomization of premise order. Also notable, the LlaMa 3.1 405B model dropped about ten percentage points in the transformation of function block (72.70%), but gained about seven percentage points in the same–different block (89.34%). Similarly, the smaller LlaMa 3.3 70B model also dropped about five percentage points in the transformation of function block (65.96%) and gained about eight percentage points in the same–different block (78.42%). The effect of increasing the number of relational premises in a problem, including irrelevant premises and manipulating the validity of the prompted conclusions were largely similar to the original results. The effect of increasing number of premises was slightly more pronounced in the replication study for GPT models (Figure A7 and Table A27). Effects of problem variations in deictic (Figure A8 and Table A28), analogy (Figure A9 and Table A29) and transformation of function blocks (Figure A10, Table A30 and Table A31) were largely the same as in the original task.

## 4. Discussion

Inspired by behavior-analytic research on the human ability for relational responding, we conducted survey of a small sample of LLMs and LRMs on a large battery of relational syllogistic reasoning problems. Our results demonstrated that both the LLMs (i.e., LlaMa 3.1 405B and 3.3 70B) and LRMs (i.e., GPT OSS 20B and 120B) in our sample generally perform well in these tasks. The LRMs reach at least eighty percent accuracy on all types of relating (including analogy and transformation of function) and reach near-maximum performance on same–different, comparison and temporal relations. The LLMs showed more variability in their performance, and while they performed at a similar level to the GPT models in the comparison, temporal, hierarchy and deictic blocks, their performance dropped markedly in the same–different, same–opposite, analogy and transformation of function blocks. This finding appears to show that the extensive post-training that LRMs go through does benefit them when it comes to solving syllogistic reasoning problems, potentially because this training involves finetuning of step-by-step reasoning (e.g., [60]; [42]; [55]; [31]). Also in line with past research on LLMs and LRMs (e.g., [38]; [10]; [13]; [31]), we observed that the larger models (i.e., LlaMa 3.1 405B and GPT OSS 120B) generally outperformed their smaller counterparts (i.e., LlaMa 3.3 70B and GPT OSS 20B, respectively).

These results appear to show that state-of-the-art systems have acquired an ability for relational responding, but they do show sensitivities to the type(s) of relations involved in the problem premises, and to variations in the problem presentation (i.e., varying the number of premises, including irrelevant premises and prompting invalid conclusions). While the effects of task variations differed between the models and between types of relations, they may reflect artifacts from the models’ training. The models’ decreased performance when invalid conclusions were prompted (except for GPT models, which were more accurate for invalid conclusions prompted in the same–opposite block) may reflect a bias for affirmative responding. Similarly, decreased performance in problems with more premises and for problems with an irrelevant premise could also reflect a lack of generalization from training. While our use of non-words likely reduces the probability that the models have seen these exact problems in training, they could still take advantage of structural overlap between these problems and those seen in training, and from the presence of cues indicating that overlap (i.e., performance is a function of overlap in problem topography, rather than generalized relational understanding). We were not able to control the training data of the models, but future research could address this. The decrease in performance on these problem variants, however, could also simply reflect the increased difficulty of these problems (i.e., problems with more relational premises have higher relational complexity, [32]), differences in prompt sensitivity or tokenization idiosyncrasies, which could also be elucidated in future research. Results of our replication study showed that the models’ performance was relatively robust against perturbations of the order of relational premises (which affected performance in prior studies, e.g., [70]), suggesting the ability to solve relational syllogisms is relatively generalized. Future research could investigate the models’ sensitivity to problem variations by other aspects of the task (e.g., multiple choice format, testing other relations) and by assessing human’s sensitivity to those same task variations.

The LLMs’ relatively low performance on analogy problems (LlaMa models performed around chance-level, while GPT models reached about 83% accuracy) somewhat nuances claims of emergent analogical reasoning in LLMs e.g., [68]; but see [41], for a counterexample). Taken together with the fact that we probed analogical reasoning in a somewhat unusual way (e.g., “A is the same as B. B is the same as C. Is A to B the same as B to C?”), one could argue that the models’ performance does not reflect truly generalized, abstract relational understanding. However, we must also note that the decrease in performance for analogy problems was most pronounced for comparative and temporal relations, where technically speaking, the model can be considered correct when it determines two identical relations (e.g., “A more than B” and “B more than C”) as not being identical, because they might not be the exact same instance of the relation (e.g., A may be five more than B, while B could be twenty more than C). Further research investigating the models’ reasoning process is needed to address this question.

A final aspect of our test of relational abilities that deserves highlighting is the transformation of function. The GPT models appeared to have little difficulty with these problems and performed at near-maximum level. LLMs’ lower performance on transformation of function trials was limited to same–different and same–opposite relations. Performance on those problems decreased with an increasing number of premises, analogous to regular same–different and same–opposite trials (no transformation of function) and may therefore reflect the effect of increasing the number of same–different and same–opposite premises, rather than a difficulty with the transformation of stimulus function itself. Both LLMs and LRMs thus appear to be capable of transforming semantic functions in a relational network, which is an important aspect of relational responding ([33]).

Further research is needed to answer the question how the models’ performance relates to that of humans. Given the limited number of studies using the RAI or its derivatives (and the different versions used therein), normative (block-level) data for human performance is unavailable. [15] ([15]) validated a version of the RAI (69 trials involving sameness, difference, opposition, comparative, temporal and analogy) and reported that adult participants scored between 80% and 90% for same–different, same–opposite, more–less and before–after trials, and that performance dropped to around 65% for analogy trials. Future research needs to collect human data for this test battery to allow for a more direct comparison of human and artificial performance, as well as the effect of problem variations on human performance.

Readers should be careful interpreting these findings, however. As we mentioned in the introduction, there are many pitfalls, such as the risk of anthropomorphizing the models and attributing human abilities and mechanisms (e.g., understanding, reasoning) to them ([43]). It is important to emphasize that output alignment (the fact that artificial systems produce seemingly human-like responses) does not a priori imply alignment of the underlying mechanisms or processes. Indeed, these models may solve these tasks in entirely different ways than humans do. The current study only investigated overall performance (whether models can solve relational syllogistic problems), but to further investigate similarities between the models’ and human reasoning, future research should study the models’ ‘reasoning’ process in more detail, and compare it with that of humans (e.g., [63]; [56]; [22]; [6]). Another limitation of the current work concerns our use of non-words in our test. We chose to do so in line with previous work in behavior-analytic literature (e.g., [15]), and argue that the use of non-words allows us to assess whether the system (human or artificial) has acquired a general ability for relational responding (i.e., if it can act as if randomly selected non-words are related, it can learn to do so with any stimulus). A downside of using non-words is that it reduces the ecological validity of our task. While we do not wish to downplay this concern, it is reassuring to know that past research has shown that performance in procedures like these (i.e., syllogistic reasoning problems that specify relations between non-words) can be considered a proxy-measure of intellectual abilities ([14]) and that they can be used to improve relational and intellectual abilities (e.g., [11]; [20]). Furthermore, we would argue that our use of non-words is intended to both (i) reduce training data contamination, and (ii) increase reliance on relational contextual cues (i.e., the relations specified in the relational premises), because the non-words are unlikely to possess functions that might control the relational responding. There are therefore theoretical reasons (see [33]) to suspect that the use of non-words may not impact ecological validity when the same cues are present in the tested syllogisms and the “real world”. Nevertheless, future research could address the question of ecological validity of these procedures in more detail.

Beyond merely assessing the relational abilities of LLMs and LRMs to get another measure of their competence, we mainly conducted this study to illustrate a broader point about using (a behavior-analytic perspective on) relational responding as a framework to evaluate artificial systems and compare them to other systems (humans or other animals). As we mentioned in the introduction, researchers in different fields of psychological research and computer science agree that relational abilities are a cornerstone of human cognition ([49]) and may even be considered a proxy-measure of general intelligence ([14]; [61]). By studying relational abilities, we can evaluate different systems without getting caught up in philosophical debates about the nature of general intelligence, yet still obtain a measure related to it. Our results showed that state-of-the-art LLMs and LRMs display the ability for relational reasoning, which is a core aspect of intelligence. Furthermore, taking RFT’s definition of relational responding ([33]), encompassing both relatively simple responding to formal stimulus relations and more complex AARR (symbolic behavior), we can assess different types or levels of relational responding across a wide range of systems (nonhuman animals, humans, artificial systems). Many animals appear to be capable of responding to formal stimulus relations (e.g., [57]), while humans were until recently assumed to be unique in their ability for AARR (e.g., [33]; [44]; [45]). Our results show that language models have developed the ability to AARR, at least as far as our test is concerned. Further research in this direction would allow us to make fair comparisons of these systems’ relational abilities and possibly provide clues about the bigger questions surrounding the level of intelligence of current state-of-the-art artificial intelligence.

Finally, RFT also emphasizes transformation of psychological function as a crucial aspect of AARR. We are not aware of any research that has directly assessed whether LLMs and LRMs have this ability. Our results provide evidence that they do. While our current procedure only probed a limited form of function transformation (i.e., transforming the meaning of nonwords), it does go beyond existing measures (e.g., [15]; [19]) that currently do not test it at all. Future research could investigate other types of transformations of function (e.g., transforming response functions: “When I say GUK, you respond with ‘yes’. GUK is opposite to TOK. I say TOK! [‘*no*’]”) or probe it in different ways (e.g., procedures like the recently developed function transformation tasks: [23]; [24]). However, there is also the broader, more philosophical question of whether artificial systems like LLMs and LRMs can develop this ability in the first place. According to RFT, the ability to AARR (and thus, the ability for transformation of function) develops through a long history of reinforcement in a verbal environment and involves abstraction of generalized patterns of relating away from formal stimulus properties involved in multiple exemplars of relations. Humans experience this learning history grounded in a multimodal sensory environment. It has been argued that experiencing many different examples of relational responding in a long and structured learning history is required for the ability to AARR (and thus, transform of function) to arise. LLMs strictly speaking only experience our world in a linguistic sense (except for more recent multimodal LLMs and LRMs, e.g., GPT o1, [54]; Gemma 3, [27]) and require orders of magnitude more training than the average human which looks much different from that of humans (e.g., [26]). One could therefore argue that they cannot develop the ability for generalized relational understanding altogether (e.g., [3]). We hope that the framework we propose here can foster progress in these debates.

## 5. Conclusions

We conducted a large-scale assessment of the relational abilities of state-of-the-art LLMs and LRMs. Our results demonstrated that these systems generally perform well in our relational syllogistic reasoning task and appear to have developed the ability for relational responding, a core characteristic of intelligence. We did observe significant differences between the models and differences in performance for different relations. Furthermore, results suggest that model size and training significantly affect performance (bigger, more intensively post-trained models perform better). Finally, sensitivities to task variations hint at the possibility that performance does not fully reflect generalized understanding, so caution is warranted when interpreting these findings.

## Figures and Tables

**Figure 1 behavsci-16-00045-f001:**
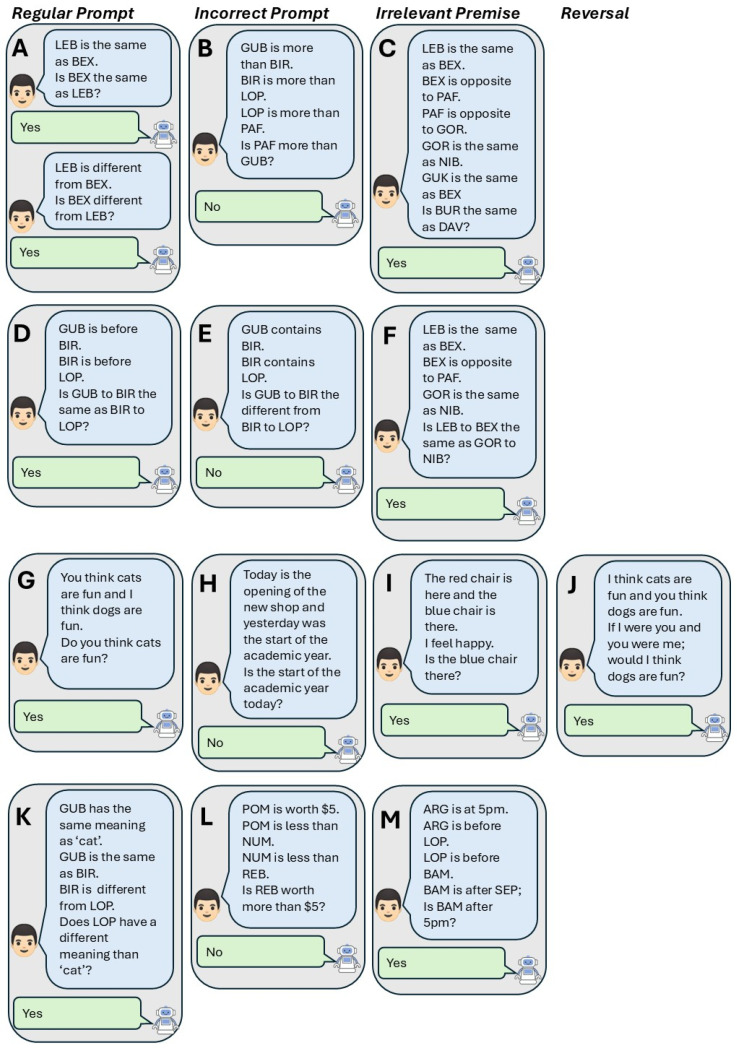
Example trials from the full task set. (**A**): Example of two one-premise problems with the correct reversal prompted. (**B**): A two-premise problem involving more than relations, with the incorrect conclusion prompted. (**C**): A four-premise problem with an irrelevant premise, all involving same–opposite relations. (**D**): An example of an analogy prompt involing temporal relations. (**E**): An analogy prompt with the incorrect conclusion prompted. (**F**): An analogy with an irrelevant premise included. (**G**–**I**): Deictic relations problems with the correct relation prompted, the incorrect relation prompted, and an irrelevant premise included, respectively. (**J**): A deictic relation problem with a reversal of the deictic dimension. (**K**–**M**): Examples of problems involving transformations of function, with the correct conclusion prompted, the incorrect conclusion prompted, and an irrelevant premise included, respectively.

**Figure 2 behavsci-16-00045-f002:**
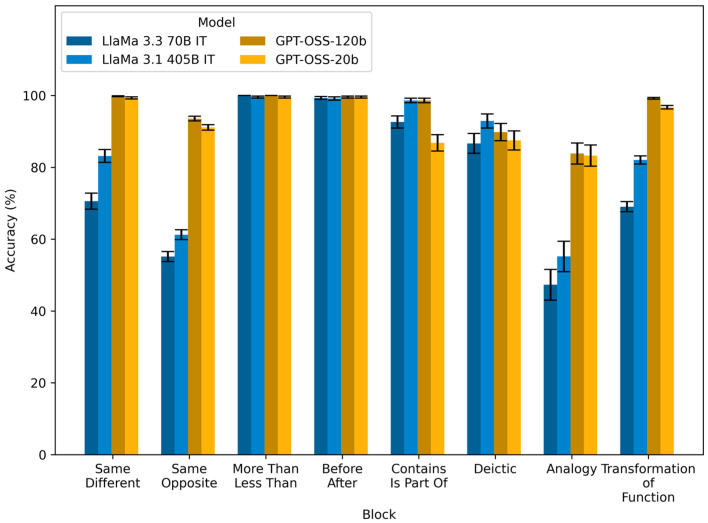
Model performance (percent accurate) grouped by the relations involved in the problems. Error bars are 95% Wilson binomial confidence intervals across items within each block.

**Figure 3 behavsci-16-00045-f003:**
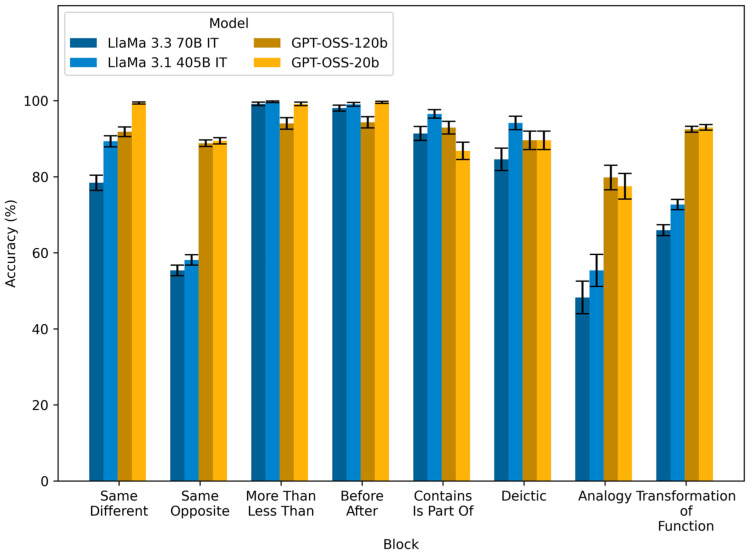
Model performance (percent accurate) in the replication study with randomized premise order, grouped by the relations involved in the problems. Error bars are 95% Wilson binomial confidence intervals across items within each block.

**Table 1 behavsci-16-00045-t001:** Overview of the tested models (and group), their training and parameter size.

Model (Group)	Training	Parameters
LLaMa 3.1 405B IT (LLM)	Pre: 15T+ multilingual, open-source text tokens. Post: Supervised Fine-Tuning, Rejection Sampling, and Direct Preference Optimization ^1^.	405B
LLaMa 3.3 70B IT (LLM)	Pre: 15T+ multilingual, open-source text tokens. Post: Supervised fine-tuning and reinforcement learning with human feedback ^1^.	70B
GPT OSS 20B (LRM)	Pre: trillions of text tokens, focus on STEM, coding, and general knowledge. Post: supervised fine-tuning and high-compute RL ^2^.	21B, 3.6 active
GPT OSS 120B (LRM)	Pre: trillions of text tokens, focus on STEM, coding, and general knowledge. Post: supervised fine-tuning and high-compute RL ^2^.	117B, 5.1 active

Note: B = billion, T = trillion. RL = Reinforcement Learning. Model names are those used in the referenced publications. ^1^
[50] ([50]). ^2^ [52] ([52]).

**Table 2 behavsci-16-00045-t002:** Model Accuracy (%) Across Blocks.

Block\Model	LlaMa 3.3 70B	LlaMa 3.3 405B	GPT OSS 20B	GPT OSS 120B
Same and Different	70.59	83.16	99.32	99.80
Same and Opposite	55.18	61.25	91.11	93.56
More Than and Less Than	100	99.58	99.58	100
Before and After	99.31	99.17	99.58	99.58
Contains and Is Part Of	92.64	98.61	86.81	98.61
Deictic	86.67	92.17	87.50	89.79
Analogy	47.31	55.19	83.27	83.85
Transformation Function	69.04	82.06	96.74	99.24

**Table 3 behavsci-16-00045-t003:** Model Accuracy (%) Across Blocks in the Replication with Randomized Premise Order.

Block\Model	LlaMa 3.3 70B	LlaMa 3.3 405B	GPT OSS 20B	GPT OSS 120B
Same and Different	78.42	89.34	99.41	91.84
Same and Opposite	55.37	58.14	89.45	88.81
More Than and Less Than	99.17	99.72	99.17	94.03
Before and After	98.06	99.03	99.58	94.31
Contains and Is Part Of	91.39	96.53	86.81	92.92
Deictic	84.58	94.17	89.58	89.58
Analogy	48.27	55.39	77.50	79.81
Transformation Function	65.96	72.70	92.99	92.48

## Data Availability

The original data presented in the study are openly available on the OSF at https://osf.io/78u36/overview?view_only=8f2df70d8ff845e9ad393d407f4c27c1 (accessed on 29 October 2025).

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
