# Peer review of "Assessing the Relational Abilities of Large Language Models and Large Reasoning Models"

_behavsci, 2025, doi:10.3390/bs16010045_

Round 1
Reviewer 1 Report
Comments and Suggestions for Authors
- Your reference list does not adhere to usual format applied for MDPI journals. Have you checked the requirements?
- Are you comparing comparable objects, LLM and LRM? They are not the same by their nature. “The evolution from LLM to LRM to LAM illustrates a broader trajectory: from understanding language, to reasoning with it, to acting on it. Each model class builds upon the previous, expanding what AI can do with human communication.”. Moreover, the title of the manuscript includes only LLM. So, it does not correspond to the body of the manuscript.
- “bigger models performed better than their smaller counterparts”. It is obvious without experiments.
- You cannot write “We assessed the intellectual abilities of state-of-the-art large language models (LLMs) and large reasoning models (LRMs)”, since you have just compared two specific models and two their versions.
- “We tested a small sample of state-of-the-art LLMs and LRMs varying in size,”. You are advised to provide the reasoning on choosing samples of LLM and LRM in the body of the manuscript. Can you make a generic conclusion if you tested a small set of LLM and LRM? How to understand “LRMs outperformed LLMs,”? You are comparing non-comparable objects. You cannot make such a conclusion like this one.
- You are advised to enumerate your tested samples in the abstract. That would add a lot of clarity to the declaration of the results of the research performed. I would observe that it is a really small set.
- You are advised to declare the group of the models (LLM, LRM) in Table 1.
Author Response
Comment 1. Your reference list does not adhere to usual format applied for MDPI journals. Have you checked the requirements?
Response 1: This is correct. As requested on the Special Issue paper call, we used the Word template provided by MDPI in the Instructions for Authors (specific for Behavioral Sciences), and formatted the reference list in line with how it was formatted in the template. We also made an APA7-formatted version (as the MDPI Instructions for Authors requested) of the manuscript available upon request.
Comment 2. Are you comparing comparable objects, LLM and LRM? They are not the same by their nature. “The evolution from LLM to LRM to LAM illustrates a broader trajectory: from understanding language, to reasoning with it, to acting on it. Each model class builds upon the previous, expanding what AI can do with human communication.”. Moreover, the title of the manuscript includes only LLM. So, it does not correspond to the body of the manuscript.
Response 2: Thank you for directing our attention to this issue. Note, however, that the comparison between the two classes of models is not our prime interest here. We included a variety of state-of-the-art models in order to get an initial exploration of possible differences but the current sample does not allow us to make general claims about them. While we do not wish to make general claims about the LLM-LRM comparison in the present paper, we do argue that the approach could be used to test a larger sample of models to do so. In choosing the sample of models, we felt like we could not exclude LRMs, given their specific training on these types of reasoning problems.
Having said this, we made several revisions to more clearly delineate the aims and scope of our work:
- We have changed the title to ‘Assessing the Relational Abilities of Large Language Models and Large Reasoning Models’, so that it corresponds to the body of the manuscript.
- In response to point (3), we have also removed the sentence hinting at the comparison between models of different sizes in the abstract (lines 17-18) because it is a trivial conclusion and not central to our current research.
- In the discussion, we have specified the models in our sample where reference to LLMs or LRMs is made (e.g., p13, line 433), or removed general claims altogether.
Comment 3. “bigger models performed better than their smaller counterparts”. It is obvious without experiments.
Response 3: Indeed. We have removed this sentence from the abstract (line 18) because we agree that it is a trivial conclusion, and because it is not a core conclusion in this study.
Comment 4. You cannot write “We assessed the intellectual abilities of state-of-the-art large language models (LLMs) and large reasoning models (LRMs)”, since you have just compared two specific models and two their versions.
Response 4: Correct, thank you for pointing this out. As also mentioned in point (2), the specific comparison of two classes of models is not our prime concern, and we agree that our current sample does not allow us to make general claims. We have further qualified this phrase in the abstract (from line 8), which now reads “We assessed the relational abilities of two state-of-the-art large language models (LLMs) and two large reasoning models (LRMs) using a new battery of several thousand syllogistic problems, similar to those used in a behavior-analytic task of relational abilities”.
Comment 5. “We tested a small sample of state-of-the-art LLMs and LRMs varying in size,”. You are advised to provide the reasoning on choosing samples of LLM and LRM in the body of the manuscript. Can you make a generic conclusion if you tested a small set of LLM and LRM? How to understand “LRMs outperformed LLMs,”? You are comparing non-comparable objects. You cannot make such a conclusion like this one.
Response 5: We have adapted the opening of to the ‘Models and Inference’ section (p. 8, paragraph 3, lines 270-275) in the following way: “We tested a sample of two state-of-the-art LLMs and two LRMs. This sample was chosen to include LLMs and LRMs varying in size, architecture and training. Although our sample was too limited to make general claims about LLMs, LRMs, or the difference in performance between the two, including a variety of models did allows us to assess relational behavior in both type of systems. Given the seminal nature of our study, we restricted the sample because of financial, environmental and practical concerns.”
Comment 6. You are advised to enumerate your tested samples in the abstract. That would add a lot of clarity to the declaration of the results of the research performed. I would observe that it is a really small set.
Response 6: We have added the sample size to the first sentence of the abstract (line 8). We agree with the reviewer that this helps contextualize the results presented thereafter.
Comment 7. You are advised to declare the group of the models (LLM, LRM) in Table 1.
Response 7: Thank you for pointing this out. We have added the model group in parentheses in Table 1 (p. 9).
Reviewer 2 Report
Comments and Suggestions for Authors
This paper proposes a benchmark of relational reasoning in LLMs and LRMs. While the study addresses a timely topic,there are still some missing considerations.
- The authors claim models "perform on par with or better than human participants" but for the specific task battery, there is a complete absence of direct human data. They cite Colbert.’s human performance on a smaller RAI but acknowledge that their own battery is "more difficult"—making the comparison invalid.
- The use of nonwords and artificially constructed syllogisms, while intended to reduce training data contamination, severely limits the ecological validity of the findings. Real world reasoning will contain a richer semantics (in both culture and in the contextualization present).
- The use of non-words reduces verbatim overlap but does not remove structural leakage.
- The abstract asserts that “bigger models performed better.” Yet the largest model tested (405 B) is only ~6× the smallest (70 B), and all models are decoder-only transformers.
Author Response
Comment 1. The authors claim models "perform on par with or better than human participants" but for the specific task battery, there is a complete absence of direct human data. They cite Colbert.’s human performance on a smaller RAI but acknowledge that their own battery is "more difficult"—making the comparison invalid.
Response 1: We agree that direct comparison to human performance is not possible in our study. We therefore no longer refer to this comparison in the abstract (line 16-17), also because it is not our primary concern. With regards to the referenced section of the discussion on page 15 (paragraph 2) , we opted to include a short section of the manuscript to the idea of comparison to human data because of the current special issue’s emphasis on ‘human-centred’ AI. We have also specified that “Further research is needed to answer the question how the models’ performance re-lates to that of humans.” (line 498-499) and removed the sentence regarding human performance in this task (line 505-507).
Comment 2. The use of nonwords and artificially constructed syllogisms, while intended to reduce training data contamination, severely limits the ecological validity of the findings. Real world reasoning will contain a richer semantics (in both culture and in the contextualization present).
Response 2: We have added a section on the limitations of the use of non-word on page 15 (paragraph 3, line 521-539) of the discussion section: “Another limitation of the current work concerns our use of non-words in our test. We chose to do so in line with previous work in behavior-analytic literature (e.g., Colbert et al., 2020), and argue that the use of non-words allows us to assess whether the system (human or artificial) has acquired a general ability for relational responding (i.e., if it can act as if randomly selected non-words are related, it can learn to do so with any stimulus). A downside of using non-words is that it reduces the ecological validity of our task. While we do not wish to downplay this concern, it is reassuring to know that past research has shown that performance in procedures like these (i.e., syllogistic reasoning problems that specify relations between non-words) can be considered a proxy-measure of intellectual abilities (Colbert et al., 2017) and that they can be used to improve relational and intellectual abilities (e.g., Cassidy et al., 2011; Dixon, 2022). Furthermore, we would argue that our use of non-words is intended to both i) reduce training data contamination, and ii) increase reliance on relational contextual cues (i.e., the relations specified in the relational premises), because the non-words are unlikely to possess functions that might control the relational responding. There are therefore theoretical reasons (see Hayes et al., 2001) to suspect that the use of non-words may not impact ecological validity when the same cues are present in the tested syllogisms and the “real world”. Nevertheless, future research could address the question of ecological validity of these procedures in more detail.”.
Comment 3. The use of non-words reduces verbatim overlap but does not remove structural leakage.
Response 3: We have added a caveat about the possibility of models taking advantage of structural leakage on p. 14 (paragraph 2, line 458-466): “While our use of non-words likely reduces the probability that the models have seen these exact problems in training, they could still take advantage of structural overlap between these problems and those seen in training, and from the presence of cues indicating that overlap (i.e., performance is a function of overlap in problem topography, rather than generalized relational understanding). We were not able to control the training data of the models, but future research could address this. The decrease in performance on these problem variants, however, could also simply reflect the in-creased difficulty of these problems (i.e., problems with more relational premises have higher relational complexity, Halford et al., 2010).”
Comment 4. The abstract asserts that “bigger models performed better.” Yet the largest model tested (405 B) is only ~6× the smallest (70 B), and all models are decoder-only transformers.
Response 4: In response to this point and a comment from another reviewer (see Point 3 by Reviewer 1), we have removed this sentence from the abstract (line 18). The comparison of model types (LLM versus LRM) or model parameters is not our primary objective here.
Reviewer 3 Report
Comments and Suggestions for Authors
The paper makes a strong and timely contribution by bringing behavior-analytic concepts of relational abilities and Relational Frame Theory into the evaluation of LLMs and LRMs. The idea of using a large battery of syllogistic problems with nonword stimuli to probe general relational abilities is original and well aligned with the behavior-analytic framing in the introduction. The introduction is generally clear, well written, and well referenced, and it does a good job of explaining why relational abilities matter for intelligence without getting bogged down in definitional debates.
However, there are a few conceptual and methodological issues that should be tightened. In the abstract and introduction, the claim that models “perform on par with or better than human participants in similar tasks” risks overinterpretation unless the human comparison is precisely matched and clearly described. The manuscript would benefit from a more careful explanation of how human and model tasks are made comparable (e.g., instructions, time limits, feedback, training, number of items) and from framing this as “comparable performance on this specific relational battery,” rather than as a general statement about intellectual ability.
The link between performance on your relational reasoning tasks and intellectual abilities or intelligence could be nuanced further. You rightly motivate relational abilities as a hallmark of intelligence, but it should be clearer that you are operationalizing one important facet, not intelligence as a whole. Similarly, when you interpret variability across relations and task variations as hinting at a lack of generalized understanding, it would help to distinguish between genuine limits in generalization and more mundane explanations (such as differences in prompt sensitivity, training data distributions, or tokenization idiosyncrasies, etc.).
Methodologically, more detail is likely needed on how the problem battery was generated, validated, and checked for unintended cues. The paper mentions randomly selected nonwords and randomized premise order, which is good, but readers will want specifics on the generation procedure, how relational structures were balanced, how validity and invalidity was defined, and how you ensured that there were no systematic artifacts that models could exploit. Because you explicitly raise concerns about training-data contamination in the introduction, you should also explain more concretely why the current battery is unlikely to have appeared in training data, and what limitations remain.
Author Response
Comment 1. In the abstract and introduction, the claim that models “perform on par with or better than human participants in similar tasks” risks overinterpretation unless the human comparison is precisely matched and clearly described. The manuscript would benefit from a more careful explanation of how human and model tasks are made comparable (e.g., instructions, time limits, feedback, training, number of items) and from framing this as “comparable performance on this specific relational battery,” rather than as a general statement about intellectual ability.
Response 1: In response to this point and comments from another reviewer, we have removed the sentences alluding to ‘comparisons to human performance’ from the abstract (line 16-17) and discussion section (p.15, paragraph 2, line 505-507). We also adapted the abstract, which now reads “We assessed the relational abilities of …” (instead of intellectual abilities; line 8) and emphasizes that “our research provides a new framework for testing a core aspect of intellectual (i.e., relational) abilities in artificial systems” (line 22).
Comment 2. The link between performance on your relational reasoning tasks and intellectual abilities or intelligence could be nuanced further. You rightly motivate relational abilities as a hallmark of intelligence, but it should be clearer that you are operationalizing one important facet, not intelligence as a whole.
Response 2: Thank you for this suggestion. As per our response to point (1), we have adapted the closing of the abstract (line 22), which now reads: “Our research provides a new framework for testing a core aspect of intellectual (i.e., relational) abilities in artificial systems (as well as humans and other animals), of which the implications and future research directions are discussed.”
Comment 3. Similarly, when you interpret variability across relations and task variations as hinting at a lack of generalized understanding, it would help to distinguish between genuine limits in generalization and more mundane explanations (such as differences in prompt sensitivity, training data distributions, or tokenization idiosyncrasies, etc.).
Response 3: We have removed this interpretation from the abstract (line 20) as we cannot provide further clarification there. In the discussion section (p. 14, paragraph 3, line 521-539), we have added a section in response to this point and a point from another reviewer (see Comment 2 by Reviewer 2): “While our use of non-words likely reduces the probability that the models have seen these exact problems in training, they could still take advantage of structural overlap between these problems and those seen in training, and from the presence of cues indicating that overlap (i.e., performance is a function of overlap in problem topography, rather than generalized relational understanding). We were not able to control the training data of the models, but future research could address this. The decrease in performance on these problem variants, however, could also simply reflect the increased difficulty of these problems (i.e., problems with more relational premises have higher relational complexity, Halford et al., 2010), differences in prompt sensitivity or tokenization idiosyncrasies, which could also be elucidated in future research.”
Comment 4. Methodologically, more detail is likely needed on how the problem battery was generated, validated, and checked for unintended cues. The paper mentions randomly selected nonwords and randomized premise order, which is good, but readers will want specifics on the generation procedure, how relational structures were balanced, how validity and invalidity was defined, and how you ensured that there were no systematic artifacts that models could exploit. Because you explicitly raise concerns about training-data contamination in the introduction, you should also explain more concretely why the current battery is unlikely to have appeared in training data, and what limitations remain.
Response 4: We included a section on the generation of syllogistic problems on p. 4 (paragraph 2, line 161-175), which reads “…The different trial numbers for the different relations were the result of the functioning of the generator function. For a number of specified relations, this function finds all valid combinations (i.e., the combination allows for valid derivation of a new relation) of the relations for a given number of premises (e.g., for two premises with sameness or difference relations, it could be sameness twice, or a combination of sameness and difference). For those combinations of relations, it then creates relational syllogistic problems with one relation per premise. Relations vary in terms of how they can be combined with other relations to allow valid derivations. For instance, two (or more) opposition relations can be specified between three (or more) relata, allowing one to derive novel transitive relations (e.g., A opposite to B and B opposite to C, so A is the same as C), whereas combinations of two or more difference relations does lead to valid derivations (if A is different from B and B is different from C, one cannot derive the relation between A and C with certainty). As a result, for some combinations of relations (e.g., same-different), fewer unique trials could be generated, leading to different numbers of trials between the relations.
We also added a few sentences describing the selection and randomization of non-words in the section describing the syllogistic reasoning problems (p. 4 and 5, line 183-188). It now reads: “…Ten variants (each with different non-words as the relata) of each unique problem were created, adding up to a total of 13640 problems (full set available on the Open Science Framework ) on which the LLMs and LRMs were assessed. To reduce the likelihood of prior experience with them in this context, the non-words used as relata were randomly selected from a pool of three-letter syllables, chosen to not be included in openly available versions of the RAI and to as much as possible not be parts of common words. Randomization of the non-words in different problems was constant across the different models. Below, we describe the main variables (beyond the relation involved) that we manipulated in the problems”
Finally, in the discussion section (p. 15, paragraph 3, line 521-539), we added a section describing limitations of the current task, with regards to the use of non-words and the need for validation.
Round 2
Reviewer 1 Report
Comments and Suggestions for Authors
Thank you for the revision.
Reviewer 2 Report
Comments and Suggestions for Authors
The authors have effectively addressed all concerns. I recommend the manuscript be accepted for publication.